# Renal Parenchyma Segmentation in Abdominal MR Images based on Cascaded Deep Neural Networks with Image and Shape Patches

**Hyeonjin Kim**[1]                                    HYUNJINKIM@SWU.AC.KR
**Helen Hong**[*1]                                     HLHONG@SWU.AC.KR
[1] *Department of Software Convergence, Seoul Womens University, Seoul, Republic of Korea*

**Dae Chul Jung**[2]                                   DAECHUL@YUHS.AC
[2] *Department of Radiology and Research Institute of Radiological Science, Yonsei University College of Medicine, Seoul, Republic of Korea*

**Kidon Chang**[3]                                     KDCHANG@YUHS.AC
[3] *Department of Urology and Urological Science Institute, Yonsei University Wonju College of Medicine, Wonju, Republic of Korea*

**Koon Ho Rha**[4]                                     KHRHA@YUHS.AC
[4] *Department of Urology, Yonsei University College of Medicine, Seoul, Republic of Korea*

**Editors:** Under Review for MIDL 2021

## Abstract

We propose an automatic segmentation method of renal parenchyma in abdominal MR images based on cascaded deep neural networks with image and shape patches. First, intensity and spacing normalization is performed in the whole MR image. Second, kidney is localized with the ensemble of 2D segmentation networks based on attention U-Net on the axial, coronal, sagittal plane. Third, signal intensity correction between each data is performed in the localized area, and renal parenchyma is segmented with the ensemble of two 3D segmentation networks based on UNet++ with image and shape patches. The average F1-score of renal parenchyma was 91.43% at left kidney, 89.30% at right kidney in F1-score, respectively.

**Keywords:** Magnetic resonance (MR), Renal parenchyma, Deep neural network, Image patch, Shape patch

## 1. Introduction

Segmentation of renal parenchyma which consists of renal cortex and renal medulla responsible for the renal function is challenging due to small region in the abdomen, blurred boundary, and similar signal intensities, location and shape with nearby organs. Furthermore, signal intensity is different for each data due to the movement caused by breathing and heartbeat when taking abdominal MR images, and the distribution of the signal intensity between the images is different even when acquired using the same device. Therefore, we propose an automatic renal parenchyma segmentation method based on cascaded deep neural network with image and shape patches in abdominal MR images.

---

* Corresponding author

## 2. Methods

To reduce the difference in signal intensity within the image due to the movement caused by breathing and heartbeat when acquired, z-score normalization and spacing normalization is performed. To localize the kidney in abdomen, ensemble of three 2D attention U-Net(Oktay et al., 2018) results at axial, coronal, sagittal plane image is performed with average voting. To segment the renal parenchyma, ensemble of 3D UNet++(Zhou et al., 2018) with image patch to consider the spatial information, and shape patch to reduce the segmentation outlier in the nearby organs by considering shape information is performed. Image patch is generated by the signal intensity correction at the localized volume. Shape patch is generated by the convolution of image patch and ensemble of three 2D segmentation results which consists of the shape of the renal parenchyma at the localized volume.

## 3. Results

Our method was evaluated on 51 abdominal T1-weighted MR images after renal partial nephrectomy acquired from GE Medical Systems-DISCOVERY MR 750 with 4.306/1.134ms TR/TE, $0.7031\sim 0.8203mm^2$ pixel size, $1.5\sim 4mm$ slice thickness and $512$x$512$x$50\sim 152$ image resolutions. 36, 9, 6 datasets were used for the training set, validation set, and test set, respectively. We compared the proposed method(Method F) to the 2D attention U-Net at axial, coronal, sagittal plane(Method A, Method B, Method C), ensemble of three 2D attention U-Net(Method D), and 3D UNet++ at the localized volume with image patch(Method E).

Table 1 and Figure 1 shows the quantitative and qualitative evaluation of renal parenchyma segmentation results. Method A shows outliers to the slices without kidney, Method B shows under-segmentation due to unclear shape at coronal plane compared to axial and sagittal planes, Method C shows outliers to the opposite side of kidney, and shows low precision and recall. Method D prevents outliers and showed better results of 15.23%p, 26.33%p, 59.53%p at the left kidney, 11.31%p, 5.65%p, 54.92%p at the right kidney in precision compared to Method A, Method B and Method C, but under-segmentation occurred at the boundary. Method E prevents under segmentation and showed better result of 4.89%p at the left kidney, 9.44%p at the right kidney in recall compared to Method D, but outlier occurred to nearby organ. Method F prevents the outlier and showed better result of 2.29%p at the left kidney, 0.18%p at the right kidney in precision compared to Method E.

Table 1: Quantitative evaluation of renal parenchyma segmentation results (%)

| | Precision | | Recall | | F1-score | |
|---|---|---|---|---|---|---|
| | left | right | left | right | left | right |
| Method A | 80.42 | 85.41 | 88.98 | 78.09 | 84.23 | 80.43 |
| Method B | 69.32 | 91.07 | 74.20 | 72.10 | 69.88 | 78.70 |
| Method C | 36.12 | 41.80 | 85.72 | 78.62 | 50.18 | 53.90 |
| Method D | 95.65 | 96.72 | 87.46 | 79.17 | 91.04 | 85.67 |
| Method E | 89.92 | 90.95 | **92.35** | **88.61** | 91.06 | 89.29 |
| Method F | **92.21** | **91.13** | 90.80 | 88.48 | **91.43** | **89.30** |

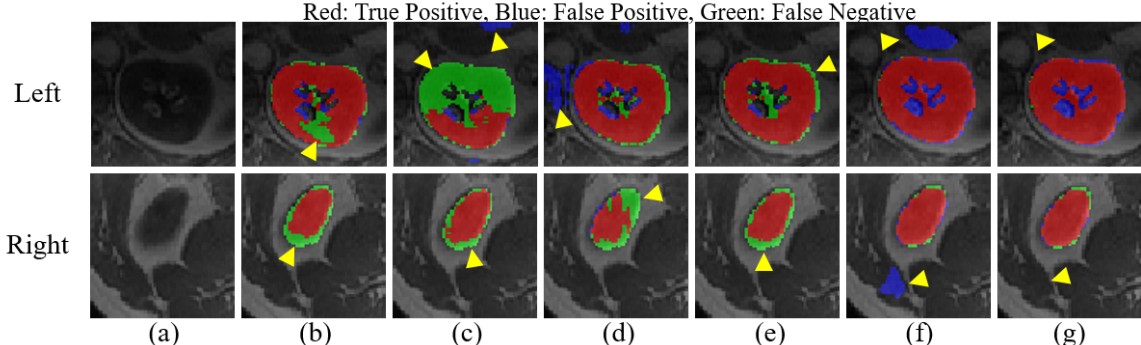

Figure 1: Qualitative evaluation of renal parenchyma segmentation results: (a) Original image, (b) Method A, (c) Method B, (d) Method C, (e) Method D, (f) Method E, (g) Method F

## 4. Conclusion

In this paper, we proposed a cascaded CNN architecture consists of renal parenchyma localization and segmentation networks with image and shape patches in abdominal MR images. Our method can be used to assess the contralateral renal hypertrophy and to predict the renal function by measuring the volume change of the renal parenchyma on MR images without radiation exposure instead of CT images, and can establish the basis for treatment after RPN.

## Acknowledgments

This work was supported Basic Science Research Program through the National Research Foundation of Korea(NRF) funded by the Ministry of Science and ICT (NRF-2019R1A2C2004746), and by Seoul R&BD Program(CY200023).

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
