# OpenReview forum: "Renal Parenchyma Segmentation in Abdominal MR Images based on Cascaded Deep Neural Networks with Image and Shape Patches"
_MIDL.io/2021/Conference/Short — Submitted to MIDL 2021_

### Official Review · Reviewer_hF5N · 2021-05-04

**Confidence:** 4
**Final Rating:** 1

**Summary:**

Summary of "Renal Parenchyma Segmentation ...": The paper proposes a segmentation pipeline for renal parenchyma in abdominal mr images using 2D localisation and 3D segmentation networks. The method is evaluated on a private in-house dataset consisting of 51 T1-weighted MR images with (presumably?) high F1-scores.

**Strengths:**

Strengths :
- The authors describe a robust and comprehensible segmentation pipeline for the task at hand.
- The short paper is well written , and most parts of the pipeline are thoroughly validated.


**Weaknesses:**

he paper type is stated as "original work" and "methodological developement" but it remains unclear what exactly the novelty is within the segmentation pipeline. The group has published a similar work at SPIE 2021, which is not referenced:

Kim, Hyeonjin, et al. "Renal parenchyma segmentation in abdominal MR images based on cascaded deep convolutional neural network with signal intensity correction." Medical Imaging 2021: Computer-Aided Diagnosis. Vol. 11597. International Society for Optics and Photonics, 2021.

The only difference to the previous work seems to be the incorporation of shape patches that have only minor impact on the segmentation accuracy (Method F vs Method E).

For an "application/validation" paper to be discussed at MIDL 2021 the clinical motivation and impact is not sufficiently described. While authors state in the conclusion that "our method can be used to assess the contralateral renal hypertrophy and to predict the renal function" it is unfortunately not further validated or described if the segmentation accuracy reached with the proposed method is good enough for the clinical task.

**Deanonymize Review:**

no

**Detailed Comments:**

To highlight the advantage of the proposed segmentation pipeline (intensity correction, ensembling, localization, segmentation) it would be helpful to report results of a strong single model 3D CNN baseline (e.g. nnUNet) on the raw images.

**Justification Of The Rating:**

 While the pipeline in general seems robust and is well described I vote for a strong reject as essential own prior work is not referenced and the methodological improvements are too incremental for a short paper at MIDL.

**Paper Type:**

validation/application paper

**Special Issue:**

no

---

### Official Review · Reviewer_TTFh · 2021-05-07

**Confidence:** 4
**Final Rating:** 3

**Summary:**

The authors propose an ensemble network for segmenting renal parenchyma from MRI signal of the kidney area. They first normalize the images, then localize the kidneys with an attention-based UNet, followed by another intensity correction, and then they segment with two 3D Unet++ networks. They compare the results of the method to using only parts of the proposed model (either the attention-based UNet or the 3D Unet++) for segmentation. Along with quantitative evaluation the authors also provide visual assessment of the results.

**Strengths:**

While using only the attention-based UNet shows higher accuracy (Method D), and using only the 3D Unet++ shows better recall (Method E), the method proposed in the paper seems to make a trade-off between the two metrics and achieves the best F1 score (Method F).
Although the F1-scores are only slightly better than for Method D and E, the proposed method achieves this while keeping a better balance between precision and recall.

**Weaknesses:**

The presentation would improve a lot by comparing the method to other available methods as well, that were not incorporated into the ensemble model. As of right now the evaluation only seems to show that the ensemble performs better than any one of its components.

**Deanonymize Review:**

no

**Detailed Comments:**

In table 1, the bold font for Recall and F1-score shows the best results, this is not true for Precision, where Method F is marked although method D clearly achieves the best precision.

**Justification Of The Rating:**

I think a more extensive evaluation of the method would prove better insight into the performance of the model, as of right now it's hard to see how well it actually performs against other methods, however the results are already visually pleasing and the trade-off between precision and recall is interesting in itself.

**Paper Type:**

methodological development

**Special Issue:**

no

---

### Meta-Review · Area_Chair_HUpM · 2021-05-09

**Recommendation:** Reject
**Confidence:** 4

**Metareview:**

While the described method itself might have some merit, both reviewer remark on limited novelty of the presented work and only marginally improved results. They also point out that own prior (conference) work is not properly cited and the evaluation is too limited. I therefore cannot recommend acceptance as this type of paper (resubmission of conference paper) is not considered an appropriate candidate for MIDL short papers.

---

### Decision · Program_Chairs · 2021-05-11

Reject